# Rheological Behavior, Textural Properties, and Antioxidant Activity of *Porphyra yezoensis* Polysaccharide

**DOI:** 10.3390/molecules30040882

**Published:** 2025-02-14

**Authors:** Chenyang Ji, Xiaoshan Long, Jingjie Wang, Bo Qi, Yang Cao, Xiao Hu

**Affiliations:** 1Key Laboratory of Aquatic Product Processing, Ministry of Agriculture and Rural Affairs, South China Sea Fisheries Research Institute, Chinese Academy of Fishery Sciences, Guangzhou 510300, China; 2Co-Innovation Center of Jiangsu Marine Bio-Industry Technology, Jiangsu Ocean University, Lianyungang 222005, China; 3Department of Nutritional Sciences, University of Connecticut, Storrs, CT 06269, USA; 4Key Laboratory of Urban Agriculture in South China, Ministry of Agriculture and Rural Affairs, Guangzhou 510640, China; 5Institute of Agricultural Economics and Information, Guangdong Academy of Agricultural Sciences, Guangzhou 510640, China; 6Sanya Tropical Fisheries Research Institute, Sanya 572000, China

**Keywords:** *Porphyra yezoensis* polysaccharide, rheological properties, gelation mechanism, calcium ions, antioxidant activities

## Abstract

*Porphyra yezoensis* has attracted much attention due to its gelling properties and bioactivity. In this study, the chemical structure of *Porphyra yezoensis* polysaccharides (PYPSs) was characterized, and the effects of concentration, temperature, pH, and calcium ion (Ca^2+^) addition on the rheological properties of PYPS were systematically investigated. Chemical composition analysis indicated that PYPS primarily contained galactose (89.76%) and sulfate (15.57%). Rheological tests demonstrated that PYPS exhibited typical pseudoplastic properties, with apparent viscosity increasing with an increasing concentration. Temperature elevation from 30 °C to 90 °C weakened the intermolecular forces and reduced the apparent viscosity, whereas neutral pH (7.0) provided an optimal electrostatic equilibrium to maintain the highest viscosity. Ca^2+^ could modulate the interactions between PYPS molecules and affect the formation of the gel network structure. When the Ca^2+^ concentration reached the optimal value of 6 mM, the calcium bridges formed between Ca^2+^ and PYPS molecules not only enhanced the rheological behavior and textural properties but also formed a smooth and well-ordered network structure, achieving the highest value of fractal dimension (D_f_ = 2.9600), though excessive Ca^2+^ disrupted this well-ordered structure. Furthermore, PYPS possessed significant scavenging ability against DPPH, ABTS, and HO• radicals, demonstrating its potential application as a natural antioxidant in functional foods.

## 1. Introduction

Marine-derived polysaccharides have been widely used in the food industry due to their excellent biological activities and diverse functional properties. As natural food additives, these polysaccharides not only have strong physiological activities, such as antioxidant [1], anti-inflammatory [2], and hypoglycemic [3], but can also be used as thickeners, gelling agents, and stabilizers to improve the rheological properties, texture, and stability of foods [4]. With the growing consumer demand for natural and functional food additives, developing novel marine polysaccharide resources and elucidating their structure–function relationships has become a crucial research direction.

*Porphyra yezoensis*, an economically important seaweed in East Asia [2], contains a variety of bioactive substances, of which *Porphyra yezoensis* polysaccharides (PYPSs) are the main active components. PYPS has a characteristic structure of →3)G4S*β*(1 → 3)G(1 → 6)G4S*α*(1 → 4)LA(1 → 6)G4S*α*(1 → unit [5]. Its molecules contain hydroxyl, carboxyl, and sulfate groups, which form a stable gel through the multiple forces of intramolecular hydrogen bonding, electrostatic interactions, and ionic bonds [6]. Previous studies have shown that the structural characteristics of PYPS (e.g., molecular weight distribution, monosaccharide composition, substituent types and distribution patterns, molecular chain conformation, etc.) are closely related to its functional properties [2,7]. In practical applications, external conditions such as concentration, temperature, pH, and ionic strength affect the gel properties and network structure formation [7,8,9]. However, the influence mechanisms of different external factors on the gel properties of PYPS have not been systematically elucidated.

The formation of polysaccharide gels involves a complex process of molecular self-assembly, and the construction of their network structure depends on the synergistic effect of multiple intermolecular forces. Polysaccharide concentration plays a fundamental role in the gel formation process as it directly affects the degree of entanglement of molecular chains and network density. Temperature can affect the gel formation process by regulating molecular chain thermal motion and hydrogen bonding. At the same time, pH changes lead to alterations in the ionization states of functional groups on polysaccharide molecules, thereby affecting the intermolecular electrostatic interactions [9]. Divalent metal ions play a key role in the formation of polysaccharide gels. Among them, calcium ions (Ca^2+^) are able to form highly ordered “egg-box structures” with carboxyl and sulfate groups on polysaccharide molecules due to their suitable ionic radius (0.99 Å) and stable divalent state. Theoretical analysis showed that an appropriate amount of Ca^2+^ could maximize the binding sites with polysaccharide molecules and form optimal gel network structures. However, an excessive amount of Ca^2+^ can lead to an excessive cross-linking of molecular chains, disrupting the molecular order arrangement of the gel and ultimately reducing the overall gel stability [6,10].

Although there have been some reports on marine polysaccharide extraction process optimization and bioactivity evaluation, the mechanism of PYPS gel formation and the external factors regulating its molecular aggregation state transition and network structure formation are still insufficient. Therefore, the present study aimed to (1) investigate the intrinsic viscosity properties of PYPS and its rheological behavior under steady-state shear; (2) study the effect of external factors such as concentration, temperature, pH, and Ca^2+^ on PYPS gelation properties; (3) comprehensively use multiple characterization methods, including texture profile analysis (TPA), scanning electron microscopy (SEM) observation, and fractal dimension (D_f_) analysis, to elucidate the molecular mechanism by which Ca^2+^ regulates the formation of the PYPS gel network structure.

## 2. Results and Discussion

### 2.1. Chemical Composition and Structural Characterization of PYPS

Chemical composition analysis indicated that PYPS primarily contained total sugar (84.01%), sulfate (15.57%), and a small amount of protein (0.42%) (Table 1). The total sugar, as the main component of PYPS, forms a viscous gel network structure through intramolecular and intermolecular hydrogen bonds, which is crucial for PYPS to maintain its functional properties under various environmental conditions [11]. The high sulfate content enhances the hydrogen supply capacity in the free radical reaction by modulating the molecular structure of polysaccharides and decreasing the hydrogen bond dissociation energy, thereby improving its antioxidant activity [12].

High-performance liquid chromatography (HPLC) results demonstrate that the monosaccharide composition of PYPS consisted of mannose (0.28%), ribose (0.35%), rhamnose (0.08%), glucuronic acid (1.85%), galacturonic acid (0.02%), glucose (0.2%), galactose (89.76%), xylose (0.79%), and fucose (6.67%) (Figure 1a). Galactose, as the most abundant monosaccharide in PYPS, together with other sugar units, contributes to the structural characteristics of sulfated heteropolysaccharide [13].

High-performance gel permeation chromatography (HPGPC) revealed that the molecular weight distribution of PYPS exhibited four characteristic peaks (Figure 1b). The overall weight-average molecular weight and number-average molecular weight were 8.05 × 10^5^ Da and 697 Da, respectively, with a polydispersity index of 1154.81. The major components were the first peak (weight-average molecular weight of 1.86 × 10^6^ Da, number-average molecular weight of 7.07 × 10^5^ Da, accounting for 41.36% of the total peak area) and the second peak (weight-average molecular weight of 7.24 × 10^4^ Da, number-average molecular weight of 4.67 × 10^3^ Da, accounting for 46.99%). These high-molecular-weight components formed a network structure through the entanglement of molecular chains, significantly affecting the rheological properties of polysaccharides [14]. The molecular weight distributions of the third peak (weight-average molecular weight of 3.86 × 10^2^ Da, number-average molecular weight of 3.76 × 10^2^ Da, accounting for 1.61%) and fourth peak (weight-average molecular weight of 1.06 × 10^2^ Da, number-average molecular weight of 78 Da, accounting for 10.04%) were in the lower range. This molecular weight distribution indicates that the functional properties of PYPS are mainly determined by the high-molecular-weight components.

As depicted in Figure 1c, PYPS showed characteristic absorption peaks in the region of 4000–400 cm^−1^. The peak at 3425 cm^−1^ was attributed to the stretching vibration of O-H. The peak at 2931 cm^−1^ corresponded to the stretching and bending vibrations of C-H [13]. The absorption at 1653 cm^−1^ was assigned to the peak of bound water [15]. In addition, the peak at 1525 cm^−1^ corresponded to the bending vibration of C-N. The weak peak at 1238 cm^−1^ corresponded to the symmetrical stretching vibration of O=S=O, indicating that PYPS contains sulfate groups. The presence of sulfate groups modifies the hydrogen-donating capacity of polysaccharides by affecting the electron cloud density distribution [12]. Moreover, the peak at 1066 cm^−1^ indicated the presence of C-O-C stretching vibration and the presence of pyranose rings in the polysaccharide structure [16]. The weak peak at 906 cm^−1^ was assigned to the 3,6-anhydro linkage C-O stretching vibration. The peaks observed at 871 cm^−1^ and 795 cm^−1^ corresponded to β- and α-type glycosidic bonds, respectively [13]. Previous studies have shown that the type of glycosidic bond affects the spatial conformation and physicochemical properties of polysaccharides, where β-glycosidic bonds are usually associated with stronger molecular rigidity while α-glycosidic bonds confer a higher flexibility to polysaccharides [3].

The ^1^H and ^13^C nuclear magnetic resonance (NMR) spectra of PYPS are presented in Figure 1d and Figure 1e, respectively. The ^1^H NMR spectrum displayed signals at 5.21, 5.08, 4.69, and 4.47 ppm, and the ^13^C NMR spectrum showed peaks at 103.17, 101.87, 101.06, and 98.01 ppm. According to previous studies [5], the ^1^H NMR signals at 5.21 and 4.69 ppm were assigned to *α*-D-galactose-4-sulfate (G4S*α*) and *β*-D-galactose-4-sulfate (G4S*β*), respectively, whereas the peaks at 5.08 and 4.47 ppm corresponded to 3,6-anhydro-*α*-L-galactose (LA) and *β*-galactose (G) units, respectively. These spectral assignments confirmed the presence of both *α* and *β* glycosidic bonds, corroborating results obtained from Fourier-transform infrared spectroscopy (FT-IR) analysis.

### 2.2. Analysis of the Rheological Properties of PYPS Gel

#### 2.2.1. Intrinsic Viscosity

Intrinsic viscosity is a fundamental parameter in characterizing the fluidic properties of polysaccharides, denoting the hydrodynamic volume occupied by a single polymer molecule [17]. As shown in Figure 2, in the range of 0.5–5 mg/mL, PYPS dilute solutions exhibit a robust linear relationship (R^2^ = 0.99), with the specific viscosity (*η_sp_*/*c*) relative to the polysaccharide. When polymer solutions are diluted, the chains segregate and move independently, which makes their interactions negligible. Their intrinsic viscosity solely depends on the dimensions of the polymer chains. Through calculation via Huggins’ empirical equation, the intrinsic viscosity of PYPS is identified as 1.11 mL/mg, with a Huggins coefficient (*k_H_*) of −0.647. The value of *k_H_* relies on the structure of polymer chains and intermolecular interactions. Moreover, a *k_H_* of −0.647 suggests that the interactions between the polymer chain segments of PYPS dilute solutions are so feeble at low concentrations that they can be disregarded [18]. Hence, the flow behavior of PYPS can be sufficiently described through its intrinsic viscosity.

#### 2.2.2. Analysis of Steady Shear Characteristics of PYPS

The flow curve of PYPS gel is shown in Figure 3a. In the range of a 0.01~1000 s^−1^ shear rate, the apparent viscosity of PYPS gels decreased with an increasing shear rate, exhibiting typical pseudoplastic rheological behavior of non-Newtonian fluids. At high shear rates, the PYPS molecular chains are oriented along the shear direction, which makes the fluid structure more uniform and leads to a decrease in apparent viscosity. This phenomenon is primarily attributed to the weakening of physical interactions between adjacent molecular chains [19]. Based on polymer solution concentrations, they can be categorized into a dilute solution, semi-dilute solution, and concentrated solution. In the dilute solution of PYPS, the distance between individual polymer chains is quite large, allowing them to move freely within the solution with relatively weak interactions. However, as the concentration increases, a dynamic “entanglement” network structure forms internally, molecular collisions become more frequent, and, consequently, a higher apparent viscosity is exhibited [1]. This phenomenon occurs due to the competition between Brownian motion and fluid dynamics. At low shear rates, Brownian motion dominates, randomizing particles and forming a substantial number of dimers and aggregates, leading to high apparent viscosity in the PYPS gel. Conversely, at high shear rates, the shear force breaks down many of these dimers and aggregates, thereby reducing the apparent viscosity of the PYPS gel [20]. Therefore, the interactions between polysaccharide molecular chains within different concentration ranges lead to gels exhibiting diverse rheological characteristics.

The pseudoplastic flow behavior of PYPS solutions can be elucidated through the power-law model. The data showed that the correlation coefficient (R^2^) for all measured samples is above 0.99, indicating that the power-law model fitting results are reliable for analyzing the flow behavior of PYPS solutions (Table 2). The flow behavior index (n) is utilized to signify the extent of differences between Newtonian and non-Newtonian fluids. If n < 1, it indicates that the polysaccharide solution acts as a pseudoplastic fluid. When n = 1, the polysaccharide solution behaves as a Newtonian fluid, its flow characteristics remaining unaffected by shear rate alterations. Conversely, when n > 1, the polysaccharide solution portrays dilatant fluid behavior [20]. All PYPS solutions have an n value of less than 1, underlining that PYPS exhibits excellent pseudoplastic fluid properties. Moreover, as the PYPS concentration increased from 0.5% to 5%, the n value decreased from 0.9671 to 0.6491, and the consistency index (*k*) increased from 0.0078 to 5.905. These findings suggest that, as the PYPS concentration increases, the degree of cross-linking and flow resistance of the independently moving molecular chains also enhance, thereby increasingly deviating from Newtonian fluid characteristics and manifesting superior pseudoplastic fluid features [19].

#### 2.2.3. The Influence of Temperature on the Apparent Viscosity of PYPS Gel

Previous studies have shown that an increase in temperature decreases the apparent viscosity of polymer solutions [21]. In our study, the effect of temperature change on the apparent viscosity of 4% PYPS further validated this phenomenon (Figure 3b). At lower temperature conditions (30 °C), the polysaccharide gel maintains a self-coiling conformation and exhibits a higher apparent viscosity. When the temperature was increased to 60 °C, electrostatic groups such as glucuronic acid and sulfate began to dissociate from the polysaccharide chain, obscuring part of the electrostatic repulsion, resulting in a gradual decrease in apparent viscosity [18]. At high temperatures (90 °C), thermal energy induces more intense molecular motions in PYPS gels, leading to an increase in intermolecular distances and a weakening of intramolecular and intermolecular forces. This thermal effect simultaneously weakens the intermolecular hydrogen bonding and electrostatic and hydrophobic interactions, which, together, result in a further decrease in the apparent viscosity of the polysaccharide gel system [22].

#### 2.2.4. The Influence of pH on the Apparent Viscosity of PYPS Gel

The pH-dependent rheological behavior of PYPS gels was investigated by adjusting the system pH using 0.1 M HCl and 0.1 M NaOH (Figure 3c). The ranking of apparent viscosities at different pHs was pH 7.0 > pH 3.0 > pH 10.0. The apparent viscosity of the PYPS gels at the native pH (7.0) was higher than the apparent viscosity under acidic or alkaline conditions. This is mainly attributed to the equilibrium of intermolecular interactions and electrostatic repulsion between sulfate groups and carboxyl groups, which maximizes the apparent viscosity of PYPS gels. Under acidic or alkaline conditions, the disruption of hydrogen bonding networks alters the chain conformations and intermolecular interactions, resulting in a decreased apparent viscosity of the gel system [7].

#### 2.2.5. The Influence of Ca^2+^ on the Apparent Viscosity of PYPS Gel

Figure 3d displays the effect of Ca^2+^ concentration on the flow curves of PYPS gels. It was found that the apparent viscosity of PYPS gels exhibited a progressive increase as the Ca^2+^ concentration increased from 0 mM to 6 mM at a constant shear rate and reached a maximum value at 6 mM. However, when the Ca^2+^ concentration was further increased to 9 mM, the apparent viscosity decreased significantly. This phenomenon is attributed to the electrostatic binding of low concentrations of Ca^2+^ with the negatively charged groups on the PYPS molecules, which diminishes the intermolecular electrostatic repulsion, thus leading to an increase in the apparent viscosity of the gels [23]. In contrast, when the Ca^2+^ concentration reached higher concentrations (9 mM), the excess Ca^2+^ would disrupt the PYPS gel network structure, altering the chain conformation and molecular interactions of the polymer, which ultimately led to a decrease in the apparent viscosity of the gel. This result is consistent with the previously reported Ca^2+^ effect on *Mesona blumes* polysaccharides [24]. These results suggest that the optimal Ca^2+^ concentration promotes cross-linking between Ca^2+^ and polysaccharides, effectively regulating the apparent viscosity of the gel system.

Based on rheological studies of these four factors, the effect of Ca^2+^ on the gel properties of PYPS is nonlinear, where gel properties do not simply correlate with an increase in Ca^2+^ concentration. Unlike predictable concentration-dependent and temperature-dependent effects or reversible pH-induced conformational changes, Ca^2+^ exhibits ion-mediated molecular modulation of the gel network, acting as a network enhancer at low to moderate concentrations while causing structural disruption at higher concentrations. Therefore, we chose different concentrations of Ca^2+^ (0, 3, 6, and 9 mM) as subsequent research subjects for detailed characterization of their texture, microstructure, and thermal stability to elucidate the formation mechanism of Ca^2+^-PYPS gels.

### 2.3. Texture and Microstructure of Ca^2+^-PYPS

#### 2.3.1. Texture Analysis of Ca^2+^-PYPS Gel

The effects of different concentrations of Ca^2+^ on the textural parameters of PYPS gels are presented in Table 3. The results show that the addition of Ca^2+^ (0–9 mM) did not significantly affect the springiness, cohesiveness, resilience, and adhesiveness of the gels (*p* > 0.05). In contrast, Ca^2+^ concentration significantly affected the hardness and chewiness of the gels. When the Ca^2+^ concentration was increased to 6 mM, the hardness and chewiness reached a maximum value of 16.13 ± 0.95 g and 5.63 ± 1.02 mJ, respectively (*p* < 0.05). However, a continued increase in Ca^2+^ concentration up to 9 mM resulted in a significant decrease in these two parameters, with hardness and chewiness falling to 13.83 ± 0.76 g and 2.43 ± 0.46 mJ, respectively (*p* < 0.05). The changes in textural properties were closely related to the interactions between Ca^2+^ and the carboxyl and sulfate groups in the PYPS molecular chain. At the optimum concentration (6 mM), Ca^2+^ enhanced the intermolecular forces through the formation of calcium bridges and improved the hardness and chewiness of the gel [23], whereas the excess Ca^2+^ (9 mM) disrupted the electrostatic equilibrium between PYPS molecules, resulting in the disruption of the gel network structure.

#### 2.3.2. Appearance and Microstructural Analysis of Ca^2+^-PYPS Gel

The appearance and microscopic characteristics of Ca^2+^-PYPS gels are shown in Figure 4a,b. SEM observation revealed that the three-dimensional network structure of the gel transitioned from rough and loose to smooth and well organized with the increase in Ca^2+^ concentration (0–6 mM). This microstructural evolution clearly demonstrated the modulating effect of Ca^2+^ on the PYPS gels. However, when Ca^2+^ was added in excess (9 mM), the gel network showed structural disruption, with a porous structure and layer separation. These structural changes originate from the affinity interaction between carboxyl groups and Ca^2+^ in PYPS, which induces the redistribution of water molecules and contributes to the formation of a homogeneous and compact structure of the gel matrix [19]. However, excessive Ca^2+^ concentration weakens the hydrogen bonding interactions in the system, leading to the disruption of the gel network structure and, ultimately, the gel surface exhibiting rough and porous characteristics.

To objectively and quantitatively describe the evolution of the microscopic structure of Ca^2+^-PYPS gels, we applied a threshold processing method for an in-depth analysis of the SEM images. By introducing fractal box count analysis, we were able to reveal the complexity of the microstructure from a quantitative perspective. In the binary threshold images, the gel network structure is represented in white or gray, while the gel pore parts appear in black. This visualization process helps in presenting the changes in the microstructure more clearly (Figure 4c). According to the box-counting method, the D_f_ values of the Ca^2+^-PYPS gels under 0, 3, 6, and 9 mM conditions were obtained as 2.8708, 2.9180, 2.9600, and 2.8752, respectively (Figure 4d). The D_f_ value reflects the complexity of the Ca^2+^-PYPS gel network structure. Theoretically, the larger the D_f_ value, the more complex the gel structure, and the more uniformly ordered the internal distribution [25]. Under all experimental conditions, the gel prepared with 6 mM of Ca^2+^ displayed the highest D_f_ value (2.9600). This result indicates that the microstructure of the Ca^2+^-PYPS gel system under this condition is more complex and orderly. Nevertheless, when the amount of Ca^2+^ added continues to increase to 9 mM, the D_f_ value slightly decreases, which is consistent with the results of rheology and texture analysis in this study.

### 2.4. Thermal Stability Analysis of Ca^2+^-PYPS Dry Gels

The thermogravimetry/derivative thermogravimetric (TG/DTG) curves can illustrate the thermal stability and intermolecular cross-link density of composite gels with different Ca^2+^ additions. Generally, the internal cross-link density of the gel network is positively correlated with thermal stability [10]. All samples showed two stages of weight loss, as shown in Figure 5. The first weight-loss phase occurs below 100 °C and is due to the evaporation and desorption of water from the sample. The second weight-loss phase occurs between 180 °C and 400 °C and is mainly caused by thermal decomposition leading to the breaking of carbon chains and hydrogen bonds, resulting in the production of carbon dioxide and water [12]. Moreover, the temperature corresponding to the maximum thermal degradation rate increased from 225 °C (0 mM Ca^2+^-PYPS) to 229 °C (6 mM Ca^2+^-PYPS) with increasing Ca^2+^ addition. Upon further increasing the Ca^2+^ addition to 9 mM, however, the temperature corresponding to the maximum thermal degradation rate decreased slightly, indicating a decrease in cross-link density. Given the minimal addition of Ca^2+^ and the small weight loss in the 180–400 °C range, the effect of Ca^2+^ interference on the thermal degradation profile of PYPS gels was negligible.

### 2.5. Antioxidant Activity Analysis

The radical scavenging effect of PYPS on DPPH, ABTS, and HO• was investigated (Figure 6). The results indicate that PYPS exhibited concentration-dependent antioxidant activity in all three assays. The DPPH radical scavenging efficiency increased from 28.60 ± 2.16% to 49.40 ± 0.29% (*p* < 0.05) as the concentration increased from 2 mg/mL to 10 mg/mL. At the same concentrations, ABTS radical scavenging efficiency reached 76.19 ± 0.88% (10 mg/mL). These findings are comparable to those of *Brasenia schreberi* water-soluble polysaccharides [26,27]. For HO• scavenging, PYPS showed a scavenging efficiency of 50.05 ± 0.25% and 59.52 ± 1.31% at concentrations of 2 and 4 mg/mL, respectively, which is higher than the reported scavenging efficiency of litchi polysaccharides under the same conditions (10% and 30%) [28]. Functional groups in the PYPS molecule interact with free radicals by forming stable non-covalent or hydrogen bonds. Among them, hydroxyl (-OH) and carboxyl (-COOH) groups can act as hydrogen donors for the direct scavenging of free radicals, while sulfate groups (-OSO_3_-) enhance electron transfer capacity due to their strong polarity, potentially enhancing the free radical scavenging efficiency [12,29]. Although the scavenging efficiency of PYPS was lower than that of vitamin C (Vc), its performance was comparable or superior to that of other natural polysaccharides reported in the literature, suggesting that PYPS has potential as a natural antioxidant agent.

### 2.6. Mechanism of Ca^2+^-PYPS Gel Formation

PYPS is essentially a polymer with a negative charge. When dissolved in an aqueous solution, its negative charge is uniformly distributed throughout the entire molecular chain of PYPS. The mutual electrostatic repulsion among the PYPS molecules helps to maintain their chains in an extended conformation. The introduction of divalent Ca^2+^ ions alters this arrangement through charge shielding, which reduces intermolecular electrostatic repulsion. This phenomenon can be explained by the widely accepted “egg-box model” [30]. According to this model, Ca^2+^ promotes the formation of single complexes and egg-box dimers, resulting in a uniform and dense gel structure.

As a high-molecular-weight sulfated polysaccharide, PYPS exhibits strong molecular chain entanglement, which provides abundant binding sites for Ca^2+^. The formation of stable calcium bridges between Ca^2+^ and the free carboxyl groups on the PYPS molecular chain enhances the integrity of the gel network structure. In addition, the carboxyl and hydroxyl groups on the PYPS molecular chains enable the formation of hydrophobic interactions and hydrogen bonds. The synergistic effect of these interactions (electrostatic interactions, hydrophobic interactions, and hydrogen bonding) overcomes the repulsive and attractive interactions in the gel system and forms a stable Ca^2+^-PYPS gel network structure. However, excess Ca^2+^ inhibits the formation of egg-box dimers, leading to the disintegration of the formed gel structure and adversely affecting the gel properties, as shown in Figure 7 [31].

## 3. Materials and Methods

### 3.1. Materials

*Porphyra yezoensis* was harvested from Jiangsu Province, China. The protein concentration assay kit was purchased from Shanghai Beyotime Biotechnology Co., Ltd. (Shanghai, China). Monosaccharide standards were purchased from Solarbio Science & Technology Co., Ltd. (Shanghai, China). 2, 2-diphenyl-1-picrylhydrazyl (DPPH) was purchased from Sigma-Aldrich (St. Louis, MO, USA). All other chemicals and reagents were of analytical grade.

### 3.2. Extraction of PYPS

The preparation of PYPS was conducted according to the reported method with slight modifications [13]. Briefly, *Porphyra yezoensis* was dried in a constant-temperature drying oven (DHG-9070A, Shanghai Yiheng Scientific Instruments Co., Ltd., Shanghai, China) at 60 °C for one week, and then crushed and sieved to obtain porphyra powder. A suspension was prepared by dispersing the powder in deionized water (1:20, *w*/*v*), and ultrasound was performed at 450 W for 30 min, followed by extraction by stirring with hot water at 90 °C for 4 h. The treated solution was filtered, concentrated by spin distillation, poured into 4 times the volume of 95% ethanol, and allowed to stand at 4 °C overnight. The precipitate was collected by centrifugation (Avanti J26XP, Beckman Coulter, Brea, CA, USA) at 8000× *g* for 15 min and redissolved in deionized water. A total of 0.1% (*w*/*v*) papain was added, a 50 °C water bath was carried out for 2.5 h, boiling water was added to inactivate the enzyme, and then the enzyme was centrifuged. The resulting supernatant was mixed with Sevage reagent (supernatant: chloroform: n-butanol = 25:5:1, *v*/*v*/*v*), dialyzed for 72 h, concentrated, and lyophilized to obtain PYPS, which was yielded at 15.05%.

### 3.3. Composition and Structural Analysis of PYPS

#### 3.3.1. Chemical Composition of PYPS

The total sugar content of PYPS was measured by the phenol–sulfuric acid method with d-glucose as the standard [32]. The protein content was determined using a BCA protein quantification kit. The sulfate base content was determined by the barium sulfate turbidimetric method, and potassium sulfate was used as the standard [33].

#### 3.3.2. Monosaccharide Composition

The monosaccharide composition of PYPS was determined by pre-column derivatization with 1-phenyl-3-methyl-5-pyrazolone (PMP). PYPS was decomposed into monosaccharides by dissolving 10 mg of PYPS in 4 mL of 2.0 mol/L TFA, vacuum sealing, and hydrolyzing at 110 °C for 8 h. The derivatization process was described previously [34]. After derivatization, the PMP-labeled polysaccharide derivatives and monosaccharide standards were analyzed on a Shimadzu LC-20AD system equipped with an Xtimate-C18 column (200 mm × 4.6 mm, 5 μm, Shimadzu Corporation, Kyoto, Japan). The parameters were as follows: mobile phase A: 0.05 mol/L potassium dihydrogen phosphate solution (pH = 6.7), mobile phase B: acetonitrile; column temperature: 30 °C; detection wavelength: 250 nm; injection volume: 20 μL; elution rate: 1.0 mL/min^−1^.

#### 3.3.3. Molecular Weight Determination

The molecular weight of PYPS was measured by high-performance gel permeation chromatography (HPGPC) [35]. The experiments were performed using a Shimadzu LC-20A instrument (Shimadzu Corporation, Kyoto, Japan) equipped with a TSK-Gel GMPWXL aqueous gel chromatography column (7.8 mm × 300 mm, Tosoh Corporation, Tokyo, Japan). PYPS was prepared as a solution at a concentration of 0.5 mg/mL^−1^, filtered through a 0.22 μm filter membrane, and injected into the sample. The analysis was carried out under the following conditions: mobile phase of 0.1 N NaNO_3_ and 0.06% NaN_3_ aqueous solution, injection volume of 20 μL, flow rate of 0.6 mL/min^−1^, and column temperature of 35 °C. The elution process was monitored by a differential refractive index detector (RID-20A, Shimadzu Corporation, Kyoto, Japan). The molecular weight calibration curve was established using dextran standards of different molecular weights (6.3, 22, 49.4, 334, and 642 kDa). Finally, the molecular weight of PYPS was calculated by LabSolutions GPC software (v5.89, Shimadzu Corporation, Kyoto, Japan) based on retention time.

#### 3.3.4. NMR

PYPS (30 mg) was fully dissolved in D_2_O (0.6 mL) and then transferred to an NMR tube [36]. The ^1^H-NMR and ^13^C-NMR spectra of the polysaccharide components were determined using an Avance III 400M NMR spectrometer (Bruker, Hamburg, Germany).

### 3.4. Preparation and Treatment of PYPS Stock Solutions

#### 3.4.1. Preparation of Stock Solutions

The PYPS powder was dissolved in deionized water at 60 °C for 2 h with continuous stirring to prepare a stock solution with a concentration range of 0.5–5% (*w*/*v*). The 4% (*w*/*v*) PYPS solution was selected for subsequent experiments and equilibrated at room temperature for 12 h before use.

#### 3.4.2. Temperature, pH, and Ca^2+^ Treatments

The effects of temperature, pH, and Ca^2+^ on the properties of PYPS were investigated separately: temperature treatments were conducted at 30, 60, and 90 °C; pH treatments were adjusted to 3, 7, and 10 by 0.1 M HCl or NaOH; and Ca^2+^ treatments were performed by the addition of CaCl₂ solution to final concentrations of 0, 3, 6, and 9 mM. All samples were tested at 25 °C for subsequent testing.

### 3.5. Rheological Properties

#### 3.5.1. Determination of Intrinsic Viscosity

PYPS solutions were prepared by serial dilution at concentrations ranging from 0.1 to 0.5 mg/mL. The intrinsic viscosity was measured using a digital viscometer (NDJ-8S, Shanghai, China) and calculated according to Huggins’ empirical formula [21]:$$\frac{\eta_{sp}}{c}=\left[\eta \right]+k_{H}{\left[\eta \right]}^{2}c$$
where *η_sp_* is defined as (*η* − *η_s_*)/*η_s_*, *η* and *ηs* are the viscosity of the PYPS stock solution and deionized water, respectively, *k_H_* is Huggins’ coefficient, [*η*] represents the intrinsic viscosity of PYPS, and *c* is the concentration of the PYPS stock solution.

#### 3.5.2. Steady Flow Measurement

Rheological experiments were performed using an MCR92 rheometer (Anton Paar, Graz, Germany) equipped with a concentric cylinder geometry (bob diameter: 26.66 mm, cup diameter: 28.91 mm). The flow curves of PYPS gels were obtained by running them over a range of shear rates from 0.01 to 1000 s^−1^. The basic rheological measurements were performed at 25 °C, and temperature effect studies were performed at 30 °C, 60 °C, and 90 °C. The data were analyzed using the power-law model [37]:$$\tau =k{\left(\gamma \right)}^{n}$$
where *τ* denotes the shear stress, *k* signifies the consistency coefficient, which directly correlates with the concentration of the PYPS gel, *γ* represents the shear rate, and *n* is the index of the power-law model.

### 3.6. Physical Properties of PYPS Gels

#### 3.6.1. TPA

The textural characteristics of the PYPS gel were determined using a texture analyzer (CT3-100-115 LFRA, Brookfield Engineering Laboratories Inc., Middleboro, MA, USA) equipped with a TA/43 cylindrical probe. The analyses were performed in TPA mode under the following conditions: speed of 1.0 mm/s, distance of 10.0 mm, and trigger force of 5.0 g. Six parameters were measured in triplicate: hardness, springiness, cohesiveness, chewiness, resilience, and adhesiveness [20].

#### 3.6.2. SEM

The microstructures of the PYPS and Ca^2+^-PYPS gels were examined using a field emission scanning electron microscope (SU8020, Hitachi High-Tech Corporation, Tokyo, Japan). The gel samples were freeze-dried, sectioned into thin slices, and then gold-sputtered under vacuum conditions. The observation was performed at an accelerating voltage of 15 kV [19].

#### 3.6.3. D_f_

SEM images were converted to 1024 × 1088-pixel eight-bit binary images using ImageJ 1.53t software (National Institutes of Health, Bethesda, MD, USA), and the complexity of the Ca^2+^-PYPS gel network was analyzed using the box-counting method. The D_f_ was calculated according to the following equations [25]:$$D=-logN_{\varepsilon }/log\varepsilon$$$$D_{f}=D+1$$
where N_ε_ represents the number of boxes in the gel network at a given scale and ε is the scaling ratio compared to the unit length and the original image. D indicates the D_f_ value of a two-dimensional space image. Therefore, it is necessary to add a dimension based on the D value to represent the actual three-dimensional gel network structure.

### 3.7. Structural Characterization of PYPS Dry Gels

#### 3.7.1. FT-IR

A Fourier transform infrared spectrometer (IR Affinity-1, Shimadzu Corporation, Kyoto, Japan) was used to determine the functional groups of PYPS and Ca^2+^-PYPS gels. Each sample (5 mg) was mixed with potassium bromide at a ratio of 1:100 and vacuum-compressed into a sheet, and the FT-IR spectra were recorded in the range of 4000–400 cm^−1^ [38].

#### 3.7.2. Thermogravimetric Analysis

The thermal stability and decomposition behavior of PYPS and Ca^2+^-PYPS gels were investigated using a thermogravimetric analyzer (STA 449 F5 Jupiter, Netzsch GmbH, Selb, Germany). A total of 5 mg of each sample was placed in an alumina crucible and measured at a heating rate of 20 °C/min over a heating range of 30 to 900 °C under a nitrogen gas flow rate of 45 mL/min [12].

### 3.8. Determination of Antioxidant Activities In Vitro

#### 3.8.1. DPPH Free Radical Scavenging Activity

DPPH radical scavenging activity was determined according to the method of Wang et al. [39] with modifications. The reaction mixture was prepared by mixing the polysaccharide solution (2–10 mg/mL) with an equal volume of DPPH solution (0.2 mmol/L), followed by dark incubation (30 min, room temperature). After the reaction period, the mixture was analyzed spectrophotometrically at λ = 517 nm. Vitamin C was used as a positive control while ethanol served as a blank control. The DPPH radical scavenging activity was calculated as follows:$$\mathrm{DPPH}\;\mathrm{radical}\;\mathrm{scavenging}\;\mathrm{activity}=\frac{\left({A_{blank}-A}_{sample}\right)}{A_{blank}}\times 100\%$$

#### 3.8.2. ABTS Radical Scavenging Activity

The scavenging potential against ABTS radicals was evaluated following the method of Tang et al. [40]. Mix 7 mmol/L ABTS solution and 2.4 mmol/L K_2_S_2_O_8_ solution in equal proportions, and leave it to react in the dark at room temperature for 24 h to form an ABTS stock solution. Then, dilute the mixture with 50% methanol solution until the absorbance at 734 nm is 0.70 ± 0.02 to form an ABTS working solution. Take 0.1 mL polysaccharide solutions of different concentrations (2–10 mg/mL) and mix thoroughly with 3.9 mL ABTS working solution, leave them to react in the dark (30 min, 25 °C), and measure the absorbance at a 734 nm wavelength. ABTS radical scavenging activity is calculated as follows:$$\mathrm{ABTS}\;\mathrm{radical}\;\mathrm{scavenging}\;\mathrm{activity}=\frac{A_{0}-\left(A_{1}-A_{2}\right)}{A_{0}}\times 100\%$$
where *A*_0_ is the absorbance of the blank control (use deionized water instead of the PYPS solution); *A*_1_ is the absorbance of the PYPS solution; and *A*_2_ is the absorbance of the 50% methanol solution (use 50% methanol instead of the PYPS solution).

#### 3.8.3. HO• Scavenging Activity

The hydroxyl radical scavenging capacity was assessed using the method of Wu et al. [13] with minor modifications. Take 2 mL of PYPS solutions of different concentrations (2–10 mg/mL) into the tube and add 1 mL of ferrous sulfate (1.5 mmol/L) and 0.7 mL of H_2_O_2_ solution (3% *w*/*v*) in sequence, mix thoroughly, and equilibrate for 10 min. Subsequently, add 0.3 mL of salicylic acid (20 mmol/L in ethanol) to the mixture, and allow it to react at 25 °C for 30 min, follow by measuring absorbance determination at 510 nm. The HO• scavenging activity is calculated as follows:$$\mathrm{HO}•\;\mathrm{scavenging}\;\mathrm{rate}=\frac{1-A_{sample}-A_{control}}{A_{blank}}\times 100\%$$
where *A_control_* is the absorbance of the control (use ethanol instead of salicylic acid); *A_sample_* is the absorbance of the PYPS solution; and *A_blank_* is the absorbance of the background group (use deionized water instead of the PYPS solution).

### 3.9. Statistical Analysis

Graphs were generated using OriginPro 2019b. Data were statistically evaluated by IBM SPSS Statistics 26.0 using one-way analysis of variance (ANOVA) combined with Duncan’s multiple comparison method. Statistical significance was defined as *p* < 0.05. Results are presented in triplicate for each experiment and are expressed as mean ± standard deviation (SD).

## 4. Conclusions

This study reveals that PYPS is rich in galactose and sulfate, with characteristic β- and α-type glycosidic bonds. Rheological studies showed that, as the concentration of PYPS increased from 0.5% to 5%, the degree of entanglement between the molecular chains strengthened and the n-value decreased from 0.9671 to 0.6491, exhibiting more obvious pseudoplastic fluid properties. When the temperature increased from 30 °C to 90 °C, the enhanced thermal energy intensified the molecular motion and weakened the intermolecular hydrogen bonding and electrostatic and hydrophobic interactions, thus reducing the apparent viscosity. At neutral pH (7.0), the electrostatic repulsion between sulfate and carboxyl groups reached equilibrium, forming the most stable gel network structure with the highest apparent viscosity. The addition of Ca^2+^ significantly influenced the gel properties of PYPS. At the optimum concentration of 6 mM, the hardness and chewiness reached the maximum values (16.13 g and 5.63 mJ, respectively), and the SEM revealed that the gel network was transformed from rough and loose to smooth and well ordered, achieving the highest D_f_ value (2.9600). However, excess Ca^2+^ (9 mM) disrupted this ordered structure. Furthermore, the presence of sulfate groups contributed to PYPS’s antioxidant potential through modulating hydrogen bond dissociation energy.

## Figures and Tables

**Figure 1 molecules-30-00882-f001:**
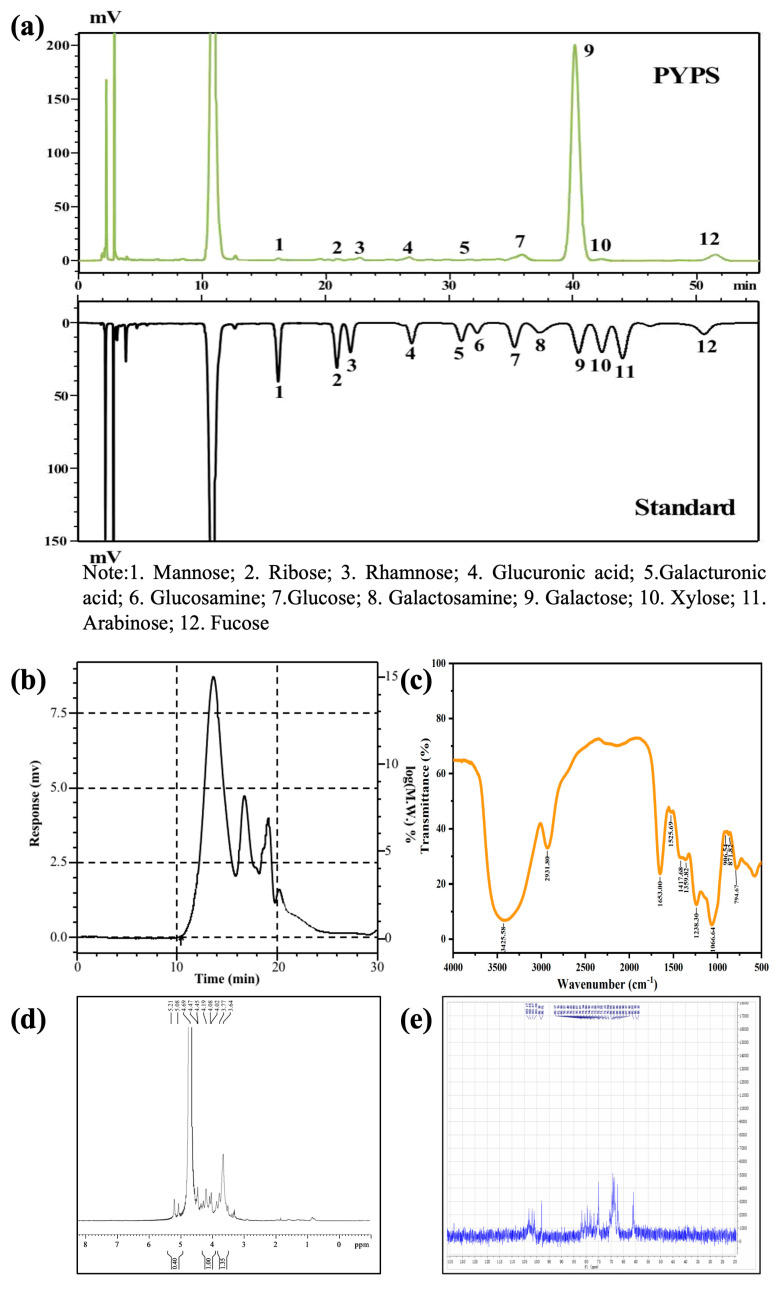
Structural characterization of PYPS. (**a**) Monosaccharide composition; (**b**) molecular weight distribution curve; (**c**) FT-IR spectrum; (**d**) ^1^H NMR; (**e**) ^13^C NMR.

**Figure 2 molecules-30-00882-f002:**
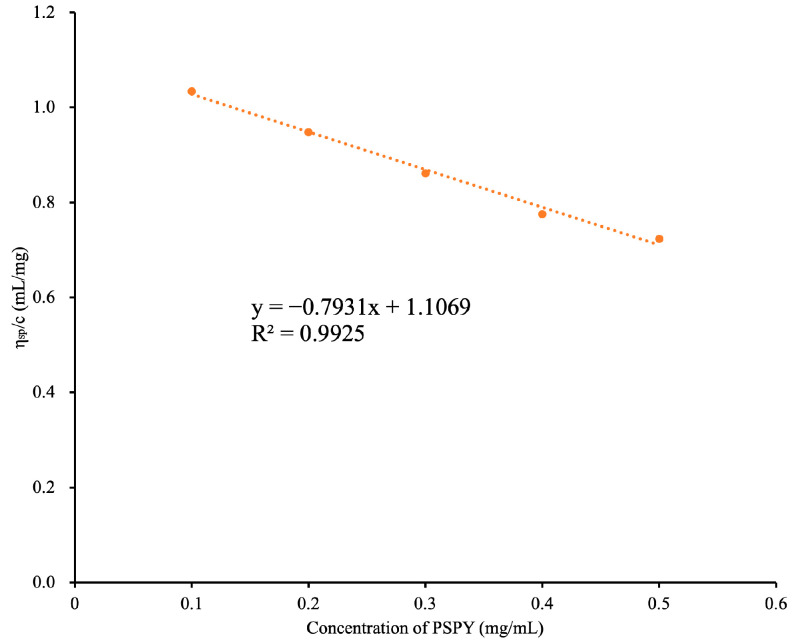
Huggins curve of dilute PYPS solution.

**Figure 3 molecules-30-00882-f003:**
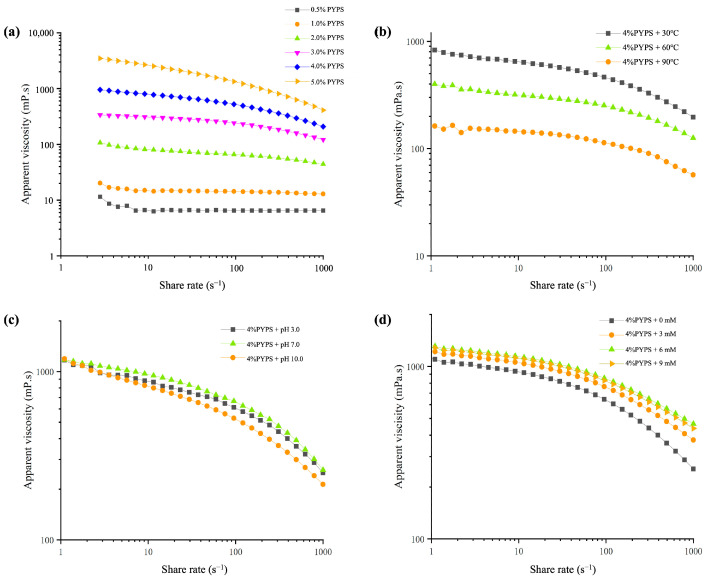
Influence of different factors on the apparent viscosity of PYPS gel. (**a**) Different concentrations; (**b**) different temperatures; (**c**) different pH values; (**d**) different amounts of Ca^2+^ addition.

**Figure 4 molecules-30-00882-f004:**
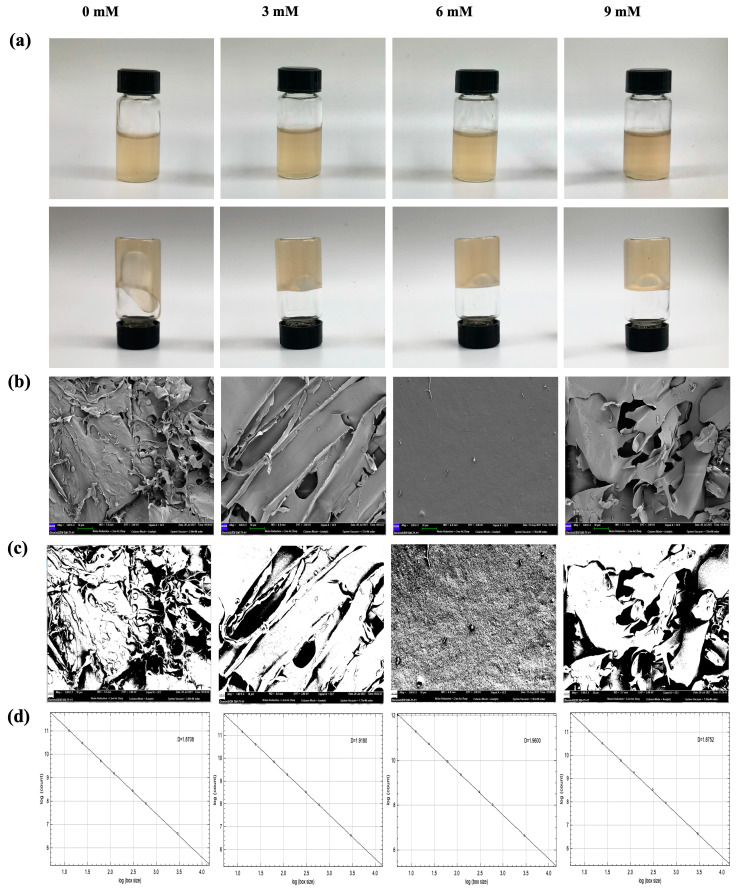
Appearance and microstructure of Ca^2+^-PYPS gels. (**a**) Appearance of Ca^2+^-PYPS gels with different concentrations; (**b**) SEM image; (**c**) binary image; (**d**) D_f_ image.

**Figure 5 molecules-30-00882-f005:**
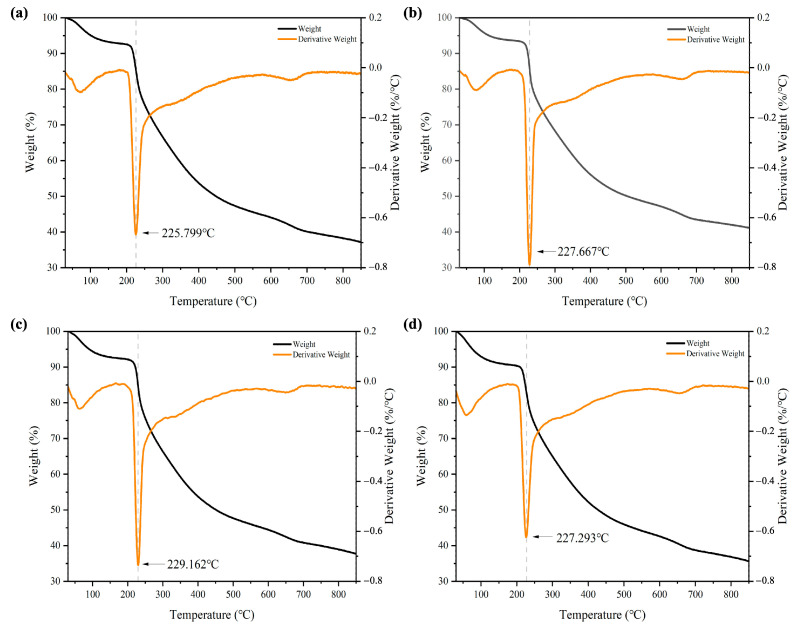
TG/DTG curves after gel freeze-drying at different Ca^2+^-PYPS additions. (**a**) 0 mM; (**b**) 3 mM; (**c**) 6 mM; (**d**) 9 mM.

**Figure 6 molecules-30-00882-f006:**
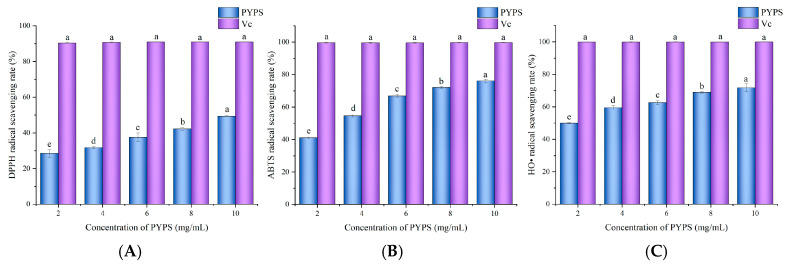
Determination of antioxidant capacity of PYPS. (**A**) DPPH radical scavenging capacity; (**B**) ABTS radical scavenging capacity; (**C**) HO• scavenging capacity. Different lowercase letters indicate significant differences (*p* < 0.05).

**Figure 7 molecules-30-00882-f007:**
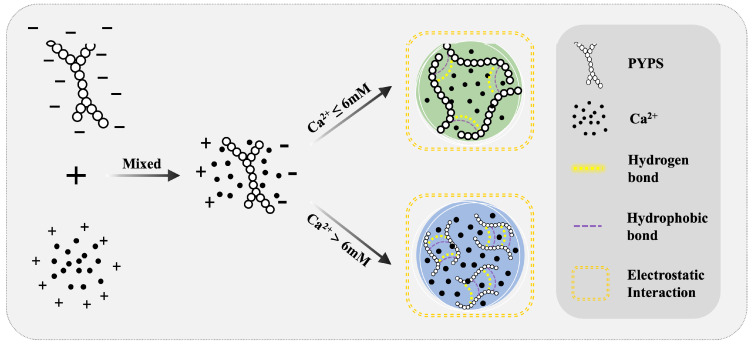
Schematic representation of the mechanism for Ca^2+^-induced Ca^2+^-PYPS gel formation.

**Table 1 molecules-30-00882-t001:** Chemical composition, monosaccharide composition, and molecular weight of PYPS.

	Composition	PYPS Sample
Chemical	Total sugar (wt.%) *	84.01
	Protein (wt.%) *	0.42
	Sulfate (wt.%) *	15.57
	Molecular weight (Da)	Overall: 8.05 × 10^5^
		Peak 1 (41.36%): 1.86 × 10^6^
		Peak 2 (46.99%): 7.24 × 10^4^
		Peak 3 (1.61%): 3.86 × 10^2^
		Peak 4 (10.04%): 1.06 × 10^2^
Monosaccharide (mol%)	Mannose	0.28
	Ribose	0.35
	Rhamnose	0.08
	Glucuronic acid	1.85
	Galacturonic acid	0.02
	Glucosamine	N.D
	Glucose	0.2
	Galactosamine	N.D
	Galactose	89.76
	Xylose	0.79
	Arabinose	N.D
	Fucose	6.67

N.D.: Not detectable or lower than the limit of quantification. * The values are presented as mean ± SD (n = 3).

**Table 2 molecules-30-00882-t002:** Determination of power-law rheological parameters for PYPS solutions of different concentrations.

	0.5%	1.0%	2.0%	3.0%	4.0%	5.0%
n	0.9671	0.9578	0.878	0.8405	0.7584	0.6491
*k*	0.0078	0.0172	0.1116	0.452	1.4042	5.905
R^2^	0.9957	0.9992	0.9994	0.9964	0.9945	0.9921

Note: n: flow behavior index; R^2^: correlation coefficient; *k*: consistency index.

**Table 3 molecules-30-00882-t003:** Textural characterization of PYPS gels with different Ca^2+^ concentrations.

Ca^2+^	Hardness (g)	Springiness (mm)	Cohesiveness (ratio)	Chewiness (mJ)	Resilience	Adhesiveness (mJ)
4% PYPS + 0 mM Ca^2+^	11.38 ± 0.75 ^a^	3.79 ± 0.95 ^a^	0.68 ± 0.05 ^a^	0.30 ± 0.08 ^a^	1.15 ± 0.03 ^a^	0.36 ± 0.03 ^a^
4% PYPS + 3 mM Ca^2+^	12.38 ± 1.44 ^ab^	3.56 ± 0.27 ^a^	0.44 ± 0.12 ^a^	4.13 ± 1.23 ^c^	1.05 ± 0.02 ^a^	0.37 ± 0.02 ^a^
4% PYPS + 6 mM Ca^2+^	16.13 ± 0.95 ^d^	3.32 ± 0.70 ^a^	0.39 ± 0.10 ^a^	5.63 ± 1.02 ^c^	0.80 ± 0.06 ^a^	0.45 ± 0.06 ^b^
4% PYPS + 9 mM Ca^2+^	13.83 ± 0.76 ^c^	2.39 ± 0.20 ^a^	0.55 ± 0.03 ^a^	2.43 ± 0.46 ^b^	0.73 ± 0.06 ^a^	0.39 ± 0.06 ^ab^

Note: Different superscript letters in the same column indicate significant differences (*p* < 0.05).

## Data Availability

Data will be made available on request.

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
