# Peer review of "Rheological Behavior, Textural Properties, and Antioxidant Activity of *Porphyra yezoensis* Polysaccharide"

_molecules, 2025, doi:10.3390/molecules30040882_

Round 1

Reviewer 1 Report

Comments and Suggestions for Authors

The author investigated the chemical structure of Porphyra yezoensis polysaccharides

(PYPS), and the effects of concentration, temperature, pH, and calciumion (Ca2+) addition on the rheological properties of PYPS. This work elucidates the structural and gel properties of PYPS, but still some questions exit in the paper.

1. None of the pictures are clear enough, please changed their pixel.

2. There are also some formatting issues, as line 31,100.

3. Some references are formatted incorrectly, as 7,14 and 37.

4. Why is there no conclusion in this paper, please add it?

5. Antioxidant activities of PYPS have studied in many papers, what’s the difference between your results with their, and should pay more attentions to gel characteristic of PYPS.

Author Response

Comments 1: None of the pictures are clear enough, please changed their pixel.

Response 1: Thank you for your suggestion. We have improved the resolution of all figures in the revised manuscript. All figures have been regenerated at higher resolution (minimum 600 dpi) to ensure better clarity and readability.

Comments 2: There are also some formatting issues, as line 31,100.

Response 2: Thank you for pointing out the formatting issues. We have corrected the format at lines 31 and 100 in the revised manuscript.

Comments 3: Some references are formatted incorrectly, as 7,14 and 37.

Response 3: Thanks for pointing this out. According to the journal's reference format requirements:

For journal articles:

Author 1, A.B.; Author 2, C.D. Title of the article. Abbreviated Journal Name Year, Volume, page range.

For book chapters:

Author 1, A.; Author 2, B. Title of the chapter. In Book Title, 2nd ed.; Editor 1, A., Editor 2, B., Eds.; Publisher: Publisher Location, Country, Year; Volume 3, pp. 154–196.

We have revised references 7, 14 and 37 accordingly:

  1. Dong, M.; Jiang, Y.; Wang, C.; Yang, Q.; Jiang, X.; Zhu, C. Determination of the extraction, physicochemical characterization, and digestibility of sulfated polysaccharides in seaweed-Porphyra haitanensis. Mar. Drugs 2020, 18,539.
  2. Guo, M.Q.; Hu, X.; Wang, C.; Ai, L. Polysaccharides: Structure and solubility. In Polysaccharides; Zhenbo, X., Ed.; IntechOpen: Rijeka, Croatia, 2017; Chapter 2, pp. 7–22.
  3. Bai, L.; Zhu, P.; Wang, W.; Wang, M. The influence of extraction pH on the chemical compositions, macromolecular characteristics, and rheological properties of polysaccharide: The case of okra polysaccharide. Food Hydrocolloids2020, 102,105586.

Comments 4: Why is there no conclusion in this paper, please add it?

Response 4: Thank you for kindly reminding us. We have added a conclusion section as follows.

This study reveals that PYPS is rich in galactose and sulfate, with characteristic β- and α-type glycosidic bonds. Rheological studies showed that as the concentration of PYPS increased from 0.5% to 5%, the degree of entanglement between the molecular chains strengthened and the n-value decreased from 0.9671 to 0.6491, exhibiting more obvious pseudoplastic fluid properties. When the temperature increased from 30 ℃ to 90 ℃, the enhanced thermal energy intensified the molecular motion and weakened the intermolecular hydrogen bonding, electrostatic and hydrophobic interactions, thus reducing the apparent viscosity. At neutral pH (7.0), the electrostatic repulsion between sulfate and carboxyl groups reached equilibrium, forming the most stable gel network structure with the highest apparent viscosity. The addition of Ca2+ significantly influenced the gel properties of PYPS. At the optimum concentration of 6 mM, the hardness and chewiness reached the maximum values (16.13 g and 5.63 mJ, respectively), and the SEM revealed that the gel network was transformed from rough and loose to smooth and well-ordered, achieving the highest Df value (2.9600). However, excess Ca2+ (9 mM) disrupted this ordered structure. Furthermore, the presence of sulfate groups contributed to PYPS's antioxidant potential through modulating hydrogen bond dissociation energy.

Comments 5: Antioxidant activities of PYPS have studied in many papers, what’s the difference between your results with their, and should pay more attentions to gel characteristic of PYPS.

Response 5: We sincerely thank the reviewer for this insightful comment. We fully agree that polysaccharide antioxidant activity studies are common in the literature, while PYPS gel characteristics deserve more investigation. Based on this suggestion, we have made the following modifications:

  1. Condensed the antioxidant activity analysis in Section 2.5, retaining only necessary findingsand adding comparisons with other literature.
  2. Expanded gel characteristics analysis in multiple sections:
  • Added moregel property analysis in the Abstract
  • Optimized the discussion of Ca2+on gel formation mechanism in Section 2.6.
  • Strengthened gel property conclusions (as described in response to Comments 4)

The revised content is as follows:

Section 2.5:

The radical scavenging effect of PYPS on DPPH, ABTS, and HO• was investigated (Figure 6). The results indicated that PYPS exhibited concentration-dependent antioxidant activity in all three assays. The DPPH radical scavenging efficiency increased from 28.60 ± 2.16% to 49.40 ± 0.29% (p < 0.05) as the concentration increased from 2 mg/mL to 10 mg/mL. At the same concentrations, ABTS radical scavenging efficiency reached 76.19 ± 0.88% (10 mg/mL). These findings are comparable to those of Brasenia schreberi water-soluble polysaccharides [26, 27]. For HO• scavenging, PYPS showed scavenging efficiency of 50.05 ± 0.25% and 59.52 ± 1.31% at concentrations of 2 and 4 mg/mL, respectively, which was higher than the reported scavenging efficiency of litchi polysaccharides under the same conditions (10% and 30%) [28]. Functional groups in the PYPS molecule interact with free radicals by forming stable non-covalent or hydrogen bonds. Among them, hydroxyl (-OH) and carboxyl (-COOH) groups can act as hydrogen donors for direct scavenging of free radicals, while sulfate groups (-OSO3-) enhance electron transfer capacity due to their strong polarity potentially enhancing the free radical scavenging efficiency [12, 29]. Although the scavenging efficiency of PYPS was lower than that of vitamin C (Vc), its performance was comparable or superior to that of other natural polysaccharides reported in the literature, suggesting that PYPS has potential as a natural antioxidant agent.

Abstract:

Porphyra yezoensis has attracted much attention due to its gelling properties and bioactivity. In this study, the chemical structure of Porphyra yezoensis polysaccharides (PYPS) was characterized, and the effects of concentration, temperature, pH, and calcium ion (Ca2+) addition on the rheological properties of PYPS were systematically investigated. Chemical composition analysis indicated that PYPS primarily contained galactose (89.76%) and sulfate (15.57%). Rheological tests demonstrated that PYPS exhibited typical pseudoplastic properties, with apparent viscosity increasing with increasing concentration. Temperature elevation from 30 ℃ to 90 ℃ weakened the intermolecular forces and reduced the apparent viscosity, whereas neutral pH (7.0) provided an optimal electrostatic equilibrium to maintain the highest viscosity. Ca2+ could modulate the interactions between PYPS molecules and affect the formation of the gel network structure. When the Ca2+ concentration reached the optimal value of 6 mM, the calcium bridges formed between Ca2+ and PYPS molecules not only enhanced the rheological behavior and textural properties, but also formed a smooth and well-ordered network structure, achieving the highest value of fractal dimension (Df = 2.9600), though excessive Ca2+ disrupted this well-ordered structure. Furthermore, PYPS possessed significant scavenging ability against DPPH, ABTS, and HO• radicals, demonstrating its potential application as a natural antioxidant in functional foods.

Section 2.6:

PYPS is essentially a polymer with a negative charge. When dissolved in an aqueous solution, its negative charge is uniformly distributed throughout the entire molecular chain of PYPS. The mutual electrostatic repulsion among the PYPS molecules helps to maintain their chains in an extended conformation. The introduction of divalent Ca2+ ions alters this arrangement through charge shielding, which reduces intermolecular electrostatic repulsion. This phenomenon can be explained by the widely accepted "egg-box model" [30]. According to this model, Ca2+ promotes the formation of single complexes and egg-box dimers, resulting in a uniform and dense gel structure.

As a high molecular weight sulfated polysaccharide, PYPS exhibits strong molecular chain entanglement, which provides abundant binding sites for Ca2+. The formation of stable calcium bridges between Ca2+ and the free carboxyl groups on the PYPS molecular chain enhances the integrity of the gel network structure. In addition, the carboxyl and hydroxyl groups on the PYPS molecular chains enable the formation of hydrophobic interactions and hydrogen bonds. The synergistic effect of these interactions (electrostatic interactions, hydrophobic interactions, and hydrogen bonding) overcomes the repulsive and attractive interactions in the gel system and forms a stable Ca2+-PYPS gel network structure. However, excess Ca2+ inhibits the formation of egg-box dimers, leading to the disintegration of the formed gel structure and adversely affecting the gel properties, as shown in Fig. 7 [31].

Conclusions:

For the modifications to the Conclusions section, please see our response to Comments 4.

We hope that these revisions adequately address the reviewers' concerns. Thank you again for your time and helpful suggestions.

Reviewer 2 Report

Comments and Suggestions for Authors

1. Lines 21-31: The abstract mentions four factors (concentration, temperature, pH, and Ca2+) that were investigated on PYPS, but the results section mainly focuses on Ca2+ effects. Please add key findings of the other three factors in the abstract.

2. Lines 20 and 48: The definition of "(Porphyra yezoensis polysaccharides) PYPS" is inconsistent between the abstract and introduction. Authors can use Porphyra yezoensis directly without defining "P. yezoensis".

3. Please verify the grammar of lines 123-134.

4. In Table 1, it seems that only one total molecular weight is reported. It is suggested that the authors could add the four peak distributions to Table 1 as well.

5. In Section 3.8.3, please clarify whether the H2O2 solution concentration (3%) is w/v or v/v.

6. Lines 534-541: It is suggested to add key findings of the other three factors in the conclusions section as well.

7. The format of references 7 and 14 needs to be consistent with the journal requirements.

Author Response

Comments 1: Lines 21-31: The abstract mentions four factors (concentration, temperature, pH, and Ca2+) that were investigated on PYPS, but the results section mainly focuses on Ca2+ effects. Please add key findings of the other three factors in the abstract.

Response 1: Thank you for your kind suggestion. We have added the key findings of the other three factors to the abstract in response to your suggestion. The revised manuscript reads as follows:

Porphyra yezoensis has attracted much attention due to its gelling properties and bioactivity. In this study, the chemical structure of Porphyra yezoensis polysaccharides (PYPS) was characterized, and the effects of concentration, temperature, pH, and calcium ion (Ca2+) addition on the rheological properties of PYPS were systematically investigated. Chemical composition analysis indicated that PYPS primarily contained galactose (89.76%) and sulfate (15.57%). Rheological tests demonstrated that PYPS exhibited typical pseudoplastic properties, with apparent viscosity increasing with increasing concentration. Temperature elevation from 30 ℃ to 90 ℃ weakened the intermolecular forces and reduced the apparent viscosity, whereas neutral pH (7.0) provided an optimal electrostatic equilibrium to maintain the highest viscosity. Ca2+ could modulate the interactions between PYPS molecules and affect the formation of the gel network structure. When the Ca2+ concentration reached the optimal value of 6 mM, the calcium bridges formed between Ca2+ and PYPS molecules not only enhanced the rheological behavior and textural properties, but also formed a smooth and well-ordered network structure, achieving the highest value of fractal dimension (Df = 2.9600), though excessive Ca2+ disrupted this well-ordered structure. Furthermore, PYPS possessed significant scavenging ability against DPPH, ABTS, and HO• radicals, demonstrating its potential application as a natural antioxidant in functional foods.

Comments 2: Lines 20 and 48: The definition of "(Porphyra yezoensis polysaccharides) PYPS" is inconsistent between the abstract and introduction. Authors can use Porphyra yezoensis directly without defining "P. yezoensis".

Response 2: Thank you for the comment. We have revised the manuscript according to your suggestion. The revised text is as follows:

Abstract:

Porphyra yezoensis has attracted much attention due to its gelling properties and bioactivity. In this study, the chemical structure of Porphyra yezoensis polysaccharides (PYPS) was characterized, and the effects of concentration, temperature, pH, and calcium ion (Ca2+) addition on the rheological properties of PYPS were systematically investigated.

Introduction:

Porphyra yezoensis, an economically important seaweed in East Asia [2], contains a variety of bioactive substances, of which Porphyra yezoensis polysaccharides (PYPS) are the main active components.

Comments 3: Please verify the grammar of lines 123-134.

Response 3: Thank you for your comment. We have revised the grammar of lines 123-134. The revision is as follows: 

As depicted in Figure 1c, PYPS showed characteristic absorption peaks in the region of 4000-400 cm-1. The peak at 3425 cm-1 was attributed to the stretching vibration of O-H. The peak at 2931 cm-1 corresponded to the stretching and bending vibrations of C-H [13]. The absorption at 1653 cm-1 was assigned to the peak of bound water [15]. In addition, the peak at 1525 cm-1 corresponded to the bending vibration of C-N. The weak peak at 1238 cm-1 corresponds to the symmetrical stretching vibration of O=S=O, indicating that PYPS contains sulfate groups. The presence of sulfate groups modifies the hydrogen donating capacity of polysaccharides by affecting the electron cloud density distribution [12]. Moreover, the peak at 1066 cm-1 indicated the presence of C-O-C stretching vibration and the presence of pyranose rings in the polysaccharide structure [16]. The weak peak at 906 cm-1 was assigned to the 3,6-anhydro linkage C-O stretching vibration. The peaks observed at 871 cm-1 and 795 cm-1 correspond to β- and α-type glycosidic bonds, respectively [13]. Previous studies have shown that the type of glycosidic bonds affects the spatial conformation and physicochemical properties of polysaccharides, where β-glycosidic bonds are usually associated with stronger molecular rigidity, while α-glycosidic bonds confer a higher flexibility to polysaccharides [3].

Comments 4: In Table 1, it seems that only one total molecular weight is reported. It is suggested that the authors could add the four peak distributions to Table 1 as well.

Response 4: Thank you for your valuable comments. According to your suggestion, we have added the molecular weight distribution information of four characteristic peaks to Table 1. Meanwhile, we have also optimized the molecular weight analysis. The revised content is as follows:

High-performance gel permeation chromatography (HPGPC) revealed that the molecular weight distribution of PYPS exhibited four characteristic peaks (Figure 1b). The overall weight-average molecular weight and number-average molecular weight were 8.05 × 105 Da and 697 Da, respectively, with a polydispersity index of 1154.81. The major components were the first peak (weight-average molecular weight of 1.86 × 106 Da, number-average molecular weight of 7.07 × 105 Da, accounting for 41.36% of the total peak area) and the second peak (weight-average molecular weight of 7.24 × 104 Da, number-average molecular weight of 4.67 × 103 Da, accounting for 46.99%). These high molecular weight components formed a network structure through the entanglement of molecular chains, significantly affecting the rheological properties of polysaccharides [14]. The molecular weight distributions of the third peak (weight-average molecular weight of 3.86 × 102 Da, number-average molecular weight of 3.76 × 102 Da, accounting for 1.61%) and fourth peak (weight-average molecular weight of 1.06 × 102 Da, number-average molecular weight of 78 Da, accounting for 10.04%) were in the lower range. This molecular weight distribution indicates that the functional properties of PYPS are mainly determined by the high molecular weight components.

Table 1. Chemical composition, monosaccharide composition, and molecular weight of PYPS.

Composition

PYPS

Chemical

Total sugar (wt. %)*

84.01

Protein (wt. %)*

0.42

Sulfate (wt. %)*

15.57

Molecular weight (Da)

Overall: 8.05 × 105

Peak 1 (41.36%): 1.86 × 106

Peak 2 (46.99%): 7.24 × 104

Peak 3 (1.61%): 3.86 × 102

Peak 4 (10.04%): 1.06 × 102

Monosaccharide

(mol %)

Mannose

0.28

Ribose

0.35

Rhamnose

0.08

Glucuronic acid

1.85

Galacturonic acid

0.02

Glucosamine

N.D

Glucose

0.2

Galactosamine

N.D

Galactose

89.76

Xylose

0.79

Arabinose

N.D

Fucose

6.67

N.D.: Not detectable or lower than the limit of quantification.

* The values are presented as mean ± SD (n = 3).

Comments 5: In Section 3.8.3, please clarify whether the H2O2 solution concentration (3%) is w/v or v/v.

Response 5: Thank you for pointing this out. We clarify that the H2O2 solution concentration (3%) used in Section 3.8.3 is expressed as w/v. The revision is as follows:

Take 2 mL of PYPS solutions of different concentrations (2-10 mg/mL) into the tube, and add 1 mL of ferrous sulfate (1.5 mmol/L) and 0.7 mL of H2O2 solution (3% w/v) in sequence, mixed thoroughly and equilibrated for 10 min.

Comments 6: Lines 534-541: It is suggested to add key findings of the other three factors in the conclusions section as well.

Response 6: Thank you for kindly reminding us. We have revised the conclusions section to include key findings of all investigated factors. The revised manuscript reads as follows:

This study reveals that PYPS is rich in galactose and sulfate, with characteristic β- and α-type glycosidic bonds. Rheological studies showed that as the concentration of PYPS increased from 0.5% to 5%, the degree of entanglement between the molecular chains strengthened and the n-value decreased from 0.9671 to 0.6491, exhibiting more obvious pseudoplastic fluid properties. When the temperature increased from 30 ℃ to 90 ℃, the enhanced thermal energy intensified the molecular motion and weakened the intermolecular hydrogen bonding, electrostatic and hydrophobic interactions, thus reducing the apparent viscosity. At neutral pH (7.0), the electrostatic repulsion between sulfate and carboxyl groups reached equilibrium, forming the most stable gel network structure with the highest apparent viscosity. The addition of Ca2+ significantly influenced the gel properties of PYPS. At the optimum concentration of 6 mM, the hardness and chewiness reached the maximum values (16.13 g and 5.63 mJ, respectively), and the SEM revealed that the gel network was transformed from rough and loose to smooth and well-ordered, achieving the highest Df value (2.9600). However, excess Ca2+ (9 mM) disrupted this ordered structure. Furthermore, the presence of sulfate groups contributed to PYPS's antioxidant potential through modulating hydrogen bond dissociation energy.

Comments 7: The format of references 7 and 14 needs to be consistent with the journal requirements.

Response 7: Thank you for your suggestion. We have revised references 7 and 14 according to the journal format requirements. The revised references are as follows:

  1. Dong, M.; Jiang, Y.; Wang, C.; Yang, Q.; Jiang, X.; Zhu, C. Determination of the extraction, physicochemical characterization, and digestibility of sulfated polysaccharides in seaweed-Porphyra haitanensis. Mar. Drugs 2020, 18, 539.
  2. Guo, M.Q.; Hu, X.; Wang, C.; Ai, L. Polysaccharides: Structure and solubility. In Polysaccharides; Zhenbo, X., Ed.; IntechOpen: Rijeka, Croatia, 2017; Chapter 2, pp. 7–22.

In addition, we have thoroughly reviewed and refined the entire manuscript. Thank you for your helpful comments and suggestions.

Reviewer 3 Report

Comments and Suggestions for Authors

In this manuscript, authors characterize Porphyra yezoensis polysaccharide employing multiple relevant techniques related to the end use of this molecule as thickener or antioxidizing agent for food products. While methods displayed here are quite “classic”, originality lies in the material studied. Experiments seem well conducted and there is certainly an interest in what authors show for a specific public. However, I have some comments and questions about the article that I think must be addressed before publication.

Overall, the manuscript is written quite well; though, graphs and figure resolution is often very low. I had trouble reading some graphs while others are outright blurry and indecipherable. This must be absolutely corrected in the final version. In addition, material and methods section is, weirdly, at the end of the article, raising a lot of questions when I read it from the beginning.

On more general remarks:

- Can authors comment on the influence of extraction methods on material properties?

- Did authors try zeta potential measurements on dilute systems to better understand ions content and pH influence?

- FT-IR comments section is, per se, interesting but I think authors should highlight the specificities of their own system (like in the last sentences of the paragraph) instead of commenting on all the “classic” peaks observed.

- Regarding salt influence on the gels, did authors tried to vary it in a “log” kind of way? (0,001 / 0,01 / 0,1 / 1 M …etc.) 

My main and most serious comments are about the rheology part:

- Since characterized materials are called “gels”, I would expect viscoelastic measurements (G’, G’’); do authors have these data?

- Measuring viscosities in continuous mode for a gelled material can affect their network (too high deformation); could author comment on that?

- I understand the use of power law to model rheological behavior, but some curves present the beginning of a Newtonian plateau. Can authors show the result of the models on the curves or use a more accurate model such as Carreau’s?

- Are there any signs of fracture or slippage during the measurements? Can the authors confirm that?

- Plotting viscosity (at a chosen shear rate) vs. salt content could be a good way to show its effect.

- Finally, there is a small correction to make in §3.5.2, as viscosity is not measured only at 25°C.

Author Response

In this manuscript, authors characterize Porphyra yezoensis polysaccharide employing multiple relevant techniques related to the end use of this molecule as thickener or antioxidizing agent for food products. While methods displayed here are quite “classic”, originality lies in the material studied. Experiments seem well conducted and there is certainly an interest in what authors show for a specific public. However, I have some comments and questions about the article that I think must be addressed before publication.

Overall, the manuscript is written quite well; though, graphs and figure resolution is often very low. I had trouble reading some graphs while others are outright blurry and indecipherable. This must be absolutely corrected in the final version. In addition, material and methods section is, weirdly, at the end of the article, raising a lot of questions when I read it from the beginning.

Response: Thank you for your constructive feedback. We have improved all figures and saved them as TIFF files at 600 dpi resolution to ensure optimal clarity. Additionally, the manuscript structure follows Molecules' template requirements, with the Materials and Methods section placed after Results.

The improved Figures are as follows:

Figure 1. Structural characterization of PYPS. (a) Monosaccharide composition; (b) Molecular weight distribution curve; (c) FT-IR spectrum; (d) 1H NMR; (e) 13C NMR.

Figure 2. Huggins curve of dilute PYPS solution.

Figure 3. Influence of different factors on the apparent viscosity of PYPS gel. (a) Different concentrations; (b) Different temperatures; (c) Different pH values; (d) Different amounts of Ca2+ addition.

Figure 4. Appearance and microstructure of Ca2+-PYPS gels. (a) Appearance of Ca2+-PYPS gels with different concentrations; (b) SEM image; (c) Binary image; (d) Df image.

Figure 5. TG/DTG curves after gel freeze-drying at different Ca2+-PYPS additions. (a) 0 mM; (b) 3 mM; (c) 6 mM; (d) 9 mM.

Figure 6. Determination of antioxidant capacity of PYPS. (A) DPPH radical scavenging capacity; (B) ABTS radical scavenging capacity; (C) HO• scavenging capacity

Figure 7. Schematic representation of the mechanism for

Ca2+-induced Ca2+-PYPS gel formation.

On more general remarks:

Comments 1: Can authors comment on the influence of extraction methods on material properties?

Response 1: Thank you for your question. In this study, we used ultrasound-assisted hot water extraction method, which is an improvement on the reported method (Wu et al., 2020) combined with ultrasonic pretreatment (450 W, 30 min) (Wang et al., 2024). Compared with the hot water extraction method reported by Wu et al. (2020), this improved method increased the yield of polysaccharides (from 10.53% to 15.05%). Meanwhile, PYPS obtained in this study had higher total sugar content (84.01% vs. 53.11%) and sulfate content (15.57% vs. 6.48%), as well as lower protein content (0.42% vs. 0.77%). These results demonstrate the effectiveness of the extraction method.

References:

Wu, Y. T.; Huo, Y. F.; Xu, L.; Xu, Y. Y.; Wang, X. L.; Zhou, T., Purification, characterization and antioxidant activity of polysaccharides from Porphyra haitanensis. Int. J. Biol. Macromol. 2020, 165, 2116-2125.

Wang, H.; Luan, F.; Shi, Y.; Yan, S.; Xin, B.; Zhang, X.; Guo, D.; Sun, J.; Zou, J., Extraction, structural features, and pharmacological effects of the polysaccharides from Porphyra yezoensis: A review. Int. J. Biol. Macromol. 2024, 279, 134745.

Comments 2: Did authors try zeta potential measurements on dilute systems to better understand ions content and pH influence?

Response 2: Thank you for your question. We did not perform zeta potential measurements. Through rheological studies, we initially investigated the effects of four factors (concentration, temperature, pH, and Ca2+) on PYPS gel properties. Based on our findings, we focused subsequent detailed characterization and mechanism description on Ca2+ effects.

The following explanation has been added to Section 2.2.5 of the manuscript:

Based on rheological studies of these four factors, the effect of Ca2+ on the gel properties of PYPS is nonlinear, where gel properties do not simply correlate with an increase in Ca2+ concentration. Unlike predictable concentration-dependent and temperature-dependent effects or reversible pH-induced conformational changes, Ca2+ exhibits ion-mediated molecular modulation of the gel network: acting as a network enhancer at low to moderate concentrations while causing structural disruption at higher concentrations. Therefore, we chose different concentrations of Ca2+ (0, 3, 6 and 9 mM) as subsequent research subjects for detailed characterization of their texture, microstructure, and thermal stability to elucidate the formation mechanism of Ca2+-PYPS gels.

Comments 3: FT-IR comments section is, per se, interesting but I think authors should highlight the specificities of their own system (like in the last sentences of the paragraph) instead of commenting on all the “classic” peaks observed.

Response 3: Thank you for your helpful suggestion. Following your comments, we have strengthened the discussion of the characteristic structure of PYPS, with particular emphasis on the absorption peaks at 1238 cm-1 (sulfate group) and 871/795 cm-1 (β-/α-glycosidic bonds).

The revisions are as follows:

The weak peak at 1238 cm-1 corresponds to the symmetrical stretching vibration of O=S=O, indicating that PYPS contains sulfate groups. The presence of sulfate groups modifies the hydrogen donating capacity of polysaccharides by affecting the electron cloud density distribution [12]. 

The peaks observed at 871 cm-1 and 795 cm-1 correspond to β- and α-type glycosidic bonds, respectively [13]. Previous studies have shown that the type of glycosidic bonds affects the spatial conformation and physicochemical properties of polysaccharides, where β-glycosidic bonds are usually associated with stronger molecular rigidity, while α-glycosidic bonds confer a higher flexibility to polysaccharides [3].

Comments 4: Regarding salt influence on the gels, did authors tried to vary it in a “log” kind of way? (0,001 / 0,01 / 0,1 / 1 M …etc.)

Response 4: Thank you for your question. The Ca2+ concentration range we selected was based on published literature (https://doi.org/10.1016/j.lwt.2021.112907). This study investigated the effect of Ca2+ concentration (0-0.012 mol/L) on gel properties.

Within our selected concentration range, Ca2+ already showed significant effects on viscosity and clearly demonstrated its action trend. Therefore, we did not use a logarithmic gradient for these experiments.

My main and most serious comments are about the rheology part:

Comments 5: Since characterized materials are called “gels”, I would expect viscoelastic measurements (G', G''); do authors have these data?

Response 5: Thank you for your question. In this study, we focused on the steady-state shear properties of PYPS under different conditions (concentration, temperature, pH and Ca2+). Through rheological tests, the samples showed obvious pseudoplastic characteristics and “gel-like” behavior, as well as rheological properties with increasing shear rate, which confirmed the typical gel properties of the samples.

Although the dynamic rheological test (G', G'') data could provide additional supporting evidence, based on the current research focus and the experimental evidence that has been obtained, we believe that the existing data adequately support the main conclusions of this paper. We will further explore the dynamic rheological properties of PYPS gels in our future studies.

Comments 6: Measuring viscosities in continuous mode for a gelled material can affect their network (too high deformation); could author comment on that?

Response 6: Thank you for your comment. The shear rate range used for the PYPS viscosity measurements was selected based on the rheological properties of polysaccharide gels and a previous study (https://doi.org/10.1080/10942912.2018.1510838). This study performed rheological analysis at concentrations ranging from 0.5-5%, using a similar range of shear rates (0.1 s-1-100 s-1) to investigate the rheological behavior of the polysaccharide system.

During our experiments, we did not observe abnormal viscosity decrease or signs of excessive deformation. The data remained stable, and samples exhibited typical shear-thinning behavior throughout the entire shear rate range, which is consistent with the rheological properties of such polysaccharide gel systems. This indicates that the selected shear rate range appropriately reflected the rheological behavior of samples within this concentration range.

Reference: https://doi.org/10.1080/10942912.2018.1510838

Comments 7: I understand the use of power law to model rheological behavior, but some curves present the beginning of a Newtonian plateau. Can authors show the result of the models on the curves or use a more accurate model such as Carreau’s?

Response 7: Thank you for your suggestion. We noticed your mention about possible Newtonian plateau in some curves. However, Figure 3 shows the rheological data obtained from rheometer measurements, not the Power Law model fitting curves. Furthermore, after careful examination of all data, we found that both the Power Law fitting in Figure 2 and the experimental measurements in Figure 3 showed continuous decrease in viscosity with increasing shear rate, without exhibiting Newtonian plateaus. Therefore, we believe that the current Power Law model adequately describes the rheological characteristics of our samples.

Comments 8: Are there any signs of fracture or slippage during the measurements? Can the authors confirm that?

Response 8: Thank you for your question. During the measurements, we did not observe any signs of fracture or slippage in our samples. In addition, our rheological data (Figure 3) show smooth curves without any unusual drops or sudden changes, which indicates that all samples maintained stable flow characteristics throughout the test.

Comments 9: Plotting viscosity (at a chosen shear rate) vs. salt content could be a good way to show its effect.

Response 9: Thank you for your suggestion. In our study, we have presented complete flow curves at different Ca2+ concentrations in Figure 3d, showing the relationship between shear rate and apparent viscosity, and thoroughly discussed the effects of Ca2+ on PYPS rheological properties in Section 2.2.5.

Considering that PYPS is a non-Newtonian fluid whose viscosity varies with the shear rate, choosing a single shear rate to plot the “Ca2+ concentration vs. viscosity” curve might not fully reflect the actual effect of Ca2+. Therefore, we maintain the complete shear rate vs. viscosity curves to show the changing pattern of viscosity under different Ca2+ concentrations and shear rates.

Comments 10: Finally, there is a small correction to make in §3.5.2, as viscosity is not measured only at 25℃.

Response 10: Thank you for pointing out this issue. Following your suggestion, we have modified the content in §3.5.2:

Original: The flow curves of PYPS gels were obtained by running them over a range of shear rates from 0.01 to 1000 s-1 at 25 ℃ and analyzed using the power-law model [37]:

Revised: The flow curves of PYPS gels were obtained by running them over a range of shear rates from 0.01 to 1000 s-1. The basic rheological measurements were performed at 25 ℃, and temperature effect studies were performed at 30 ℃, 60 ℃, and 90 ℃. The data were analyzed using the power-law model [37]:

Finally, we sincerely thank you for your professional review and valuable suggestions. We hope these revisions have adequately addressed your concerns. Thank you again for your precious time!

Round 2

Reviewer 1 Report

Comments and Suggestions for Authors

The paper have been well revised.

Reviewer 3 Report

Comments and Suggestions for Authors

This manuscript version has seen, in my opinion, great improvements. Graphs are now clearer, and my main questions were answered. I appreciate the details given to extraction method, IR part, and the additions to article "context" as well as choices made for experiments. While I regret the use of such limited Ca concentrations, I understand it relates better to the literature, as well as the way rheology is used for "practical" results, again, in accordance with existing papers methodology (I would advice doing differently for future measurements, as there is often a gap between rheology focused papers and application oriented ones). Same for G', G'' measurements, but this could indeed be a future work. In consequence, I think this article's quality is more than good for publication as it is right now.